# Statistically Robust Sparse High-order Interaction Model

**Diptesh Das**[*1]    **Ichiro Takeuchi**[2,3]    **Koji Tsuda**[†1,3,4]

[1]Department of Computational Biology and Medical Sciences, The University of Tokyo, Japan.
[2]Department of Mechanical Systems Engineering, Nagoya University, Japan.
[3]RIKEN Center for Advanced Intelligence Project, Japan.
[4]Center for Basic Research on Materials, National Institute for Materials Science, Japan.

## Abstract

Deep learning models often achieve high accuracy but lack interpretability, making them unsuitable for critical applications such as medical diagnosis, biomolecule design, criminal justice, etc. The Sparse High-Order Interaction Model (SHIM) addresses this limitation by providing both transparency and predictive reliability. However, real-world data often contain outliers, which can distort model performance. To overcome this, we propose Huberized-SHIM, an extension of SHIM that integrates Huber loss-based robust regression to mitigate the impact of outliers. We introduce a homotopy-based exact regularization path algorithm and a novel tree-pruning criterion to efficiently manage interaction complexity. Additionally, we incorporate the conformal prediction framework to enhance statistical reliability. Empirical evaluations on synthetic and real-world datasets demonstrate the superior robustness and accuracy of Huberized-SHIM in high-stakes decision-making contexts.

## 1 INTRODUCTION

While deep neural networks and other black-box models often achieve high predictive accuracy, their lack of interpretability makes them less reliable [Rudin, 2019]. Consequently, in critical applications like medical diagnosis, biomolecule design, criminal justice, etc. where transparency is essential for decision-making, models with greater interpretability and high accuracy are preferred. The sparse high-order interaction model (SHIM) [Suzumura et al., 2017, Das et al., 2019, 2022, 2024] offers both interpretability and strong predictive performance, making it

---

[*]Corresponding author: diptesh.das@edu.k.u-tokyo.ac.jp
[†]Corresponding author: tsuda@k.u-tokyo.ac.jp

a suitable choice for such tasks. Considering a regression problem of $m$ original covariates $z_1, \ldots, z_m$ and response $y$, an example SHIM up to $4^{th}$ order interactions can be written as

$$y = \beta_1 z_2 + \beta_2 z_3 + \beta_3 z_2 z_5 + \beta_4 z_1 z_3 z_4 z_6,$$

where $\beta$'s are the regression coefficients. A SHIM has significant practical applications. For example, identifying complex genotypic traits related to HIV-1 drug resistance [Saigo et al., 2007, Das et al., 2022, 2024] where a combination of multiple mutations, along with certain key single mutations provides the most accurate representation of the intricate biological mechanisms underlying drug resistance [Vivet-Boudou et al., 2006, Iversen et al., 1996, Rhee et al., 2006] or recognizing patterns of epistasis where the interdependence of mutations is crucial for understanding the relationship between genotype and phenotype [Poelwijk et al., 2019, Fannjiang et al., 2022]. Key protein characteristics, such as folding, biochemical function, and evolvability, emerge from a network of cooperative energetic interactions among amino acid residues. Identifying epistasis plays a significant role in reconstructing phylogenetic trees and assessing the evolutionary potential of antibiotic resistance genes and viruses. Additionally, in protein engineering and directed evolution, insights into epistatic structures can aid in selecting optimal templates, targeting mutations in highly epistatic regions, and identifying cooperative units for DNA shuffling experiments. Another example is criminal recidivism prediction that aims to determine the likelihood of an individual being arrested within a specific period after their release from jail or prison [Larson et al., 2016, Angelino et al., 2018]. In such cases, where predictions directly impact human lives, a model that is both highly accurate and interpretable is essential for ensuring fairness and transparency in decision-making [Rudin, 2019].

However, real-world data are often contaminated with outliers and the presence of outliers can highly influence the data-driven modeling. For example, Reichel [2025] recently studied that how the presence of a single outlier can cause

*Accepted for the $8^{th}$ Workshop on Tractable Probabilistic Modeling at UAI (TPM 2025).*

an otherwise insignificant coefficient to appear statistically significant in finite-sample inference. To counter this generally robust regression model [Wilcox, 1996] is used which instead of of automatically removing outliers, helps mitigate their impact [Tsukurimichi et al., 2022]. Robust regression modifies the loss function to downweight the effect of extreme residuals and a common choice is Huber loss, which combines squared loss for small residuals and absolute loss for large residuals [Huber, 1964, Owen, 2007, Huber and Ronchetti, 2011]. Huber loss-based regression models have been successfully used in biology [Deng et al., 2021, Deutelmoser et al., 2021], medicine [Normolle, 1993], medical diagnosis [Karim et al., 2023], finance [He et al., 2021, Pervez and Ali, 2024], and others [Das, 2023, Korgialas and Kotropoulos, 2023].

In this paper we extend SHIM and proposed Huberized-SHIM to counter the effect of outliers so that it can be used reliably even in the presence of outliers. We provided a homotopy-based exact regularization path following algorithm to compute the entire regularization path of Huberized-SHIM. We derived a novel tree-pruning criteria essential for fitting a SHIM which is otherwise intractable due the combinatorial explosion of the interaction terms. Furthermore, we integrated conformal prediction framework to demonstrate the statistical efficiency of proposed Huberized-SHIM over SHIM. We demonstrated the computational and statistical efficiency of the proposed framework using synthetic and real world data.

## 2 PROBLEM STATEMENT

Consider a regression problem with a response vector $y \in \mathbb{R}^n$ and $m$ original covariate vectors $z_1, \ldots, z_m$, where $z_j \in \mathbb{R}^n$ and $j \in [m]$. A high-order interaction model up to the $d^{\text{th}}$ order is then written as follows:

$$y = \sum_{j_1 \in [m]} \theta_{j_1} z_{j_1} + \sum_{\substack{(j_1, j_2) \in [m] \times [m] \\ j_1 \neq j_2}} \theta_{j_1, j_2} z_{j_1} z_{j_2} + \cdots$$
$$+ \sum_{\substack{(j_1, \ldots, j_d) \in [m]^d \\ j_1 \neq \ldots \neq j_d}} \theta_{j_1, \ldots, j_d} z_{j_1} \cdots z_{j_d} + \epsilon, \quad (1)$$

where $z_{j_1} \cdots z_{j_d}$ is the element-wise product, scalar $\theta$ represents the coefficient and $\epsilon$ is the noise. In this study, we mainly consider each element of the original covariate vector $z_j \in \{0, 1\}^n$. However, our model is equally applicable to covariate vectors defined in the domain $[0, 1]^n$. To simplify the notation, it is convenient to write the high-order interaction model in (1) using the following matrix of concatenated vectors of all high-order interactions:

$$X = [\underbrace{z_1, \ldots, z_m}_{1^{\text{st}} \text{ order}}, \cdots, \underbrace{z_1 \ldots z_d, \ldots, z_{m-d+1} \ldots z_m}_{d^{\text{th}} \text{ order}}] \in \mathbb{R}^{n \times p},$$

where $p := \sum_{\kappa=1}^{d} \binom{m}{\kappa}$, considering up to $d^{th}$ order interactions. Similarly, the coefficient vector associated with

all possible high-order interaction terms can be written as follows:

$$\beta := [\underbrace{\theta_1, \ldots, \theta_m}_{1^{\text{st}} \text{ order}}, \cdots, \underbrace{\theta_{1, \ldots, d}, \ldots, \theta_{m-d+1, \ldots, m}}_{d^{\text{th}} \text{ order}}]^\top \in \mathbb{R}^p.$$

The high-order interaction model (1) is then simply written as a linear model $y = X\beta + \epsilon$. Unfortunately, $p$ can be prohibitively large unless both $m$ and $d$ are fairly small. In the SHIM, we consider a sparse estimation of a high-order interaction model. An example of a SHIM is as follows:

$$y = \theta_2 z_2 + \theta_3 z_3 + \theta_{2,6} z_2 z_6 + \theta_{1,2,4,6} z_1 z_2 z_4 z_6 + \epsilon.$$

## 3 PROPOSED METHOD

We propose a *homotopy-mining* method to compute the exact path of Huberized-SHIM. The homotopy method refers to an optimization framework for solving a sequence of parameterized optimization problems. In robust (Huberized) SHIM we solve the following optimization problem:

$$\beta(\lambda) \in \underset{\beta \in \mathbb{R}^p}{\arg\min} \sum_{i=1}^{n} L(r_i(\lambda)) + \lambda ||\beta||_1 + \frac{1}{2}\alpha ||\beta||_2^2, \quad (2)$$

where $r_i(\lambda) = y_i - X_i^\top \beta(\lambda)$ is the residual, $\lambda$ and $\alpha$ are the regularization parameters of $\ell_1$ and $\ell_2$ penalty terms. The loss $L(\cdot)$ is the Huber loss:

$$L(r_i(\lambda)) = \begin{cases} \frac{1}{2} r_i^2(\lambda), & \text{if} |r_i(\lambda)| \leq \delta, \\ \delta |r_i(\lambda)| - \frac{\delta^2}{2}, & \text{otherwise.} \end{cases}$$

where $\delta \geq 0$ is a hyperparameter. We further define two new parameters $a$ and $s$ as stated in (3) to redefine the optimization problem (2):

$$a_i(\lambda) = \begin{cases} 1, & \text{if} |r_i(\lambda)| \leq \delta, \\ 0, & \text{otherwise.} \end{cases}, \quad \text{and}$$

$$s(r_i(\lambda)) = \begin{cases} \pm 1, & \text{if } r_i(\lambda) \neq 0, \\ 0, & \text{otherwise.} \end{cases} \quad (3)$$

Now, we can rewrite the loss in (2):

$$\sum_{i=1}^{n} L(r_i(\lambda)) = \sum_{i=1}^{n} \frac{1}{2} a_i(\lambda) r_i^2(\lambda) + \delta \sum_{i=1}^{n} (1 - a_i(\lambda)) |r_i(\lambda)|.$$

**Optimality conditions.** At optima we can write

$$X^\top h(\lambda) - \alpha \beta(\lambda) = \lambda s(\beta(\lambda)), \quad (4)$$

where $h(\lambda) = a(\lambda) \odot r(\lambda) + \delta(1 - a(\lambda)) \odot s(r(\lambda))$ and $\forall \ell \in [p]$,

$$s_\ell(\beta(\lambda)) = \begin{cases} \{-1, +1\}, & \text{if } \beta_\ell(\lambda) \neq 0, \\ [-1, +1], & \text{if } \beta_\ell(\lambda) = 0, \end{cases} \quad (5)$$

and $\odot$ represents element-wise vector product. Let's define the active set

$$\mathcal{A}_\lambda = \{\ell \in [p] : |x_\ell^\top h(\lambda) - \alpha\beta_\ell| = \lambda\}, \qquad (6)$$

where $[p] = \{1, 2, \ldots, p\}$ is the set of indices of all possible interaction terms of a SHIM, and the non-active set can be defined as the complement of the active set:

$$\mathcal{A}_\lambda^c = \{[p] \setminus \mathcal{A}_\lambda\}.$$

The solutions $\beta(\lambda)$ of (2) at different values of $\lambda$ is called the regularization path and the exact regularization path $(\lambda \mapsto \beta(\lambda))$ of the Huberized-SHIM can be shown to be piecewise linear as stated in Proposition 1.

**Proposition 1.** *If $\beta(\lambda)$'s have the same sign between two points $\lambda_1$ and $\lambda_2$, that is $sign(\beta(\lambda_1))=sign(\beta(\lambda_2))=sign(\beta(\lambda)), \forall\lambda \in [\lambda_1, \lambda_2)$, then $\mathcal{A}_\lambda = \mathcal{A}_{\lambda_1}$. Furthermore, assuming that $X_{\mathcal{A}_\lambda}^\top X_{\mathcal{A}_\lambda}$ is invertible and there is no "knot-crossing" for any instance $i \in [n]$ such that the values of $a(\lambda)$ and $(1 - a(\lambda))s(r(\lambda))$ remain the same for all $\lambda \in [\lambda_1, \lambda_2]$, we have the linear relations*

$$\beta_{\mathcal{A}_\lambda}(\lambda_2) = \beta_{\mathcal{A}_\lambda}(\lambda_1) + (\lambda_1 - \lambda_2)\psi_{\mathcal{A}_\lambda}(\lambda),$$
$$\lambda_2 s\big(\beta_{\mathcal{A}_\lambda^c}(\lambda_2)\big) = \lambda_1 s\big(\beta_{\mathcal{A}_\lambda^c}(\lambda_1)\big) + (\lambda_1 - \lambda_2)\gamma_{\mathcal{A}_\lambda^c}(\lambda),$$

*where the direction vectors $\psi$ and $\gamma$ are defined as*

$$\psi_{\mathcal{A}_\lambda}(\lambda) = \Big(\alpha\mathbb{I}_{|\mathcal{A}_\lambda|} + X_{\mathcal{A}_\lambda}^\top(a(\lambda) \odot X_{\mathcal{A}_\lambda})\Big)^{-1}s\big(\beta_{\mathcal{A}_\lambda}(\lambda)\big),$$
$$\gamma_{\mathcal{A}_\lambda^c}(\lambda) = -X_{\mathcal{A}_\lambda^c}^\top(a(\lambda) \odot X_{\mathcal{A}_\lambda})\psi_{\mathcal{A}_\lambda}(\lambda). \qquad (7)$$

All the proofs have been deferred to the appendix unless specified. For simplicity, we will define the step size $\Delta = (\lambda_1 - \lambda_2) > 0$. Therefore, according to Proposition 1 it is possible to design an algorithm to compute the exact $\lambda$-path of Huberized SHIM using a homotopy algorithm that exploits the linearity of the path between each two consecutive transition points of direction $(\psi, \gamma)$ changes. A homotopy algorithm of robust-SHIM sequentially tracks and updates the sign and the active set of the optimal solutions, and the parameter vectors $s(r)$ and $a$ depending on the signs and values of each component of the residual vector $r$. At any two consecutive steps represented by $\lambda_t$ and $\lambda_{t+1}$, where $t$ is an index of the transition points (kinks) of the $\lambda$-path, either of the following three events occurs:

- (Addition): a zero variable becomes non-zero, that is,

  $$\exists\ell \in \mathcal{A}_{\lambda_t}^c, \quad \text{s.t.} \quad |x_\ell^\top h(\lambda_{t+1})| = \lambda_{t+1}, \quad \text{or,}$$

- (Deletion): a non-zero variable becomes zero, that is,

  $$\exists\ell \in \mathcal{A}_{\lambda_t}, \quad \text{s.t.} \quad \beta_\ell(\lambda_t) \neq 0, \quad \text{but} \quad \beta_\ell(\lambda_{t+1}) = 0 \quad \text{or,}$$

- (Knot-crossing): a residual $r_i$ hits a Huberized knot point and the value of $a_i$ changes, that is,

  $$\exists i \in [n], \quad \text{s.t.} \quad |y_i - X_{i, \mathcal{A}(\lambda_t)}\beta_{\mathcal{A}_{\lambda_t}}(\lambda_{t+1})| = \delta$$

Overall, the next change in the direction vectors occur at $\lambda_{t+1} = \lambda_t + \Delta$, such that

$$\Delta = \min\Big(\Delta_1(\ell_1^*), \Delta_2(\ell_2^*), \Delta_3(i^*)\Big) \qquad (8)$$

where

$$\ell_1^* = \underset{\ell \in \mathcal{A}_{\lambda_t}^c}{\arg\min} \Delta_1(\ell),$$
$$\ell_2^* = \underset{\ell \in \mathcal{A}_{\lambda_t}}{\arg\min} \Delta_2(\ell),$$
$$i^* = \underset{i \in [n]}{\arg\min} \Delta_3(i), \quad \text{and}$$

$$\Delta_1(\ell) = \left(\frac{(x_\ell \mp x_k)^\top h(\lambda_t) \pm \alpha\beta_k(\lambda_t)}{(x_\ell \mp x_k)^\top(a(\lambda_t) \odot v(\lambda_t)) \mp \alpha\psi_k(\lambda_t)}\right)_{++},$$

$$\Delta_2(\ell) = \left(-\frac{\beta_\ell(\lambda_t)}{\psi_\ell(\lambda_t)}\right)_{++},$$

$$\Delta_3(i) = \left\{\min\left(\left(\frac{r_i(\lambda_t) - \delta}{v_i(\lambda_t)}\right)_{++}, \left(\frac{r_i(\lambda_t) + \delta}{v_i(\lambda_t)}\right)_{++}\right)\right\},$$

for any $k \in \mathcal{A}_{\lambda_t}$, and we defined $v(\lambda) = X_{\mathcal{A}_\lambda}\psi_{\mathcal{A}_\lambda}(\lambda)$. Here, we use the convention that for any $g \in \mathbb{R}$, $(g)_{++} = g$, if $g > 0$ and $\infty$ otherwise. However, naively (by simply minimizing over all possible interaction terms) determining the step size of inclusion $(\Delta_1(\ell_1^*))$ will be intractable for the SHIM type problem. In SHIM, the search space grows exponentially due to the combinatorial effect of high-order interaction terms. Therefore, fitting of a SHIM is non-trivial and a SHIM model will have a significantly large number of parameters to be considered unless both number of features $(m)$ and the order of interactions $(d)$ are very small. Several algorithms for fitting a sparse high-order interaction model have been proposed in the literature [Tsuda, 2007, Saigo et al., 2009, Nakagawa et al., 2016, Das et al., 2022, 2024]. A common approach adopted in these existing works is to exploit the hierarchical structure of high-order interaction features. In other words, a tree structure of interaction terms (patterns) is considered and a branch-and-bound strategy is employed in order to avoid handling all the exponentially increasing number of high-order interaction features. Hence, we need an efficient computational method to make the computation practically feasible. In the following section, we present an efficient tree pruning strategy where each node of the tree represents an interaction term. The basic idea of tree pruning is that we construct a tree of interaction terms in a 'progressive manner'. That is, we keep track of the current minimum step size of inclusion up to the construction of $\ell^{th}$ pattern as we construct the tree progressively, and prune a large part of the tree if some bound condition fails (Lemma 1).

**Lemma 1.** *For any given node $\ell$, if $\Delta_1(\ell_1^\dagger)$ is the current minimum step size, that is,*

$$\ell_1^\dagger = \underset{j \in \{1,2,\ldots,\ell\} \cap \mathcal{A}_{\lambda_t^c}}{\arg\min} \Delta_1(j),$$

*then $\forall \ell' \supset \ell, \Delta_1(\ell') \geq \Delta_1(\ell_1^\dagger)$ if*

$$b_\ell(w(\lambda_t)) + \Delta_1(\ell_1^\dagger) b_\ell(u(\lambda_t)) + b_\ell(\kappa(\lambda_t)) \quad (9)$$
$$< |\bar\rho_k(\lambda_t)| - \Delta_1(\ell_1^\dagger)|\bar\eta_k(\lambda_t)| - |\theta_k(\lambda_t)|,$$

where $w(\lambda_t) = a(\lambda_t) \odot r(\lambda_t)$, $u(\lambda_t) = a(\lambda_t) \odot v(\lambda_t)$, $\kappa(\lambda_t) = \delta(1 - a(\lambda_t)) \odot s(r(\lambda_t))$, $\bar\rho_k(\lambda_t) = \rho_k(\lambda_t) - \alpha\beta_k(\lambda_t)$, $\bar\eta_k(\lambda_t) = \eta_k(\lambda_t) + \alpha\psi_k(\lambda_t)$, $\rho_k(\lambda_t) = x_k^\top w(\lambda_t)$, $\eta_k(\lambda_t) = x_k^\top u(\lambda_t)$, $\theta_k(\lambda_t) = x_k^\top \kappa(\lambda_t)$, and for a vector $g \in \mathbb{R}^n$ we defined

$$b_\ell(g) := \max\left\{ \sum_{g_i > 0} |g_i| x_{i\ell}, \sum_{g_i < 0} |g_i| x_{i\ell} \right\}.$$

The Lemma 1 essentially states that if for any node $\ell$ the condition in (9) is satisfied, then one can safely ignore the subtree with $\ell$ as the root node, thereby dramatically improving the computational efficiency. The complete algorithm to compute the entire exact regularization path of Huberized-SHIM has been provided in the appendix.

## 4 RESULTS AND DISCUSSION

We evaluated our proposed method using both synthetic and real-world data. For details about the data please see the appendix. To demonstrate the statistical efficiency we used inductive conformal prediction [Papadopoulos et al., 2002, Angelopoulos and Bates, 2021] and reported the mean and standard deviation of the prediction set lengths ('length') and coverage ('cov') along with coefficient of determination ($R^2$) in Table 1, 2, 3. For all experiments, we considered a coverage guarantee of 90%, that is, significance level = 0.1. Please see the appendix for a brief introduction to conformal prediction. We consider 'clean' data and then gradually added 'outliers' to demonstrate the difference. The results show the 'mean (standard deviation)' of 10 independent runs in the order of Huberized-SHIM / SHIM. We consider interaction terms up to $3^{rd}-$order in both cases. To demonstrate the efficacy of tree pruning we have shown the mean fraction of node counts averaged over 10 independent runs in Figure 1. The 'fraction of nodes' for a specific 'maximum depth' represents the number of nodes traversed divided by the total number of possible combinations of interaction terms using that maximum depth. The results are shown for three different sparsity levels (0.4, 0.6, 0.8). Tree pruning is more effective at high data sparsity level which is obvious considering the tree antimonotonicity property (see tree definition in appendix for details).

Table 1: Results using synthetic SHIM data.

|  | Clean | One outlier | Five outliers | Ten outliers |
|---|---|---|---|---|
| length | 5.53 (1.77) / 7.72 (1.28) | 5.72 (1.81) / 35.84 (11.41) | 6.09 (2.17) 94.38 (21.18) | 6.49 (2.17) / 137.26 (23.57) |
| cov | 0.89 (0.05) / 0.90 (0.04) | 0.91 (0.03) / 0.92 (0.03) | 0.91 (0.03) 0.89 (0.04) | 0.89 (0.06) 0.89 (0.03) |
| $R^2$ | 0.94 (0.02) / 0.95 (0.02) | 0.94 (0.04) / 0.01 (0.54) | 0.93 (0.04) -5.78 (2.29) | 0.93 (0.04) / -12.11 (3.63) |

Table 2: Results using Fluorescence data (fitness='red').

|  | Clean | One outlier | Five outliers | Ten outliers |
|---|---|---|---|---|
| length | 0.58 (0.08) / 0.58 (0.08) | 0.62 (0.07) / 0.85 (0.15) | 1.17 (0.48) 1.68 (0.34) | 1.85 (0.39) / 2.30 (0.42) |
| cov | 0.91 (0.03) / 0.91 (0.03) | 0.91 (0.02) / 0.90 (0.04) | 0.91 (0.06) 0.92 (0.05) | 0.92 (0.04) 0.91 (0.06) |
| $R^2$ | 0.65 (0.10) / 0.65 (0.10) | 0.58 (0.10) / 0.26 (0.30) | -0.24 (0.88) -1.37 (0.94) | -2.35 (1.85) / -4.39 (2.70) |

Table 3: Results using Compas data.

|  | Clean | One outlier | Five outliers | Ten outliers |
|---|---|---|---|---|
| length | 9.15 (0.63) / 9.48 (0.77) | 9.14 (0.64) / 9.50 (0.72) | 9.14 (0.68) 12.13 (1.68) | 9.33 (0.67) / 16.11 (2.84) |
| cov | 0.93 (0.03) / 0.93(0.02) | 0.93 (0.03) / 0.91 (0.03) | 0.93 (0.03) 0.93(0.02) | 0.93 (0.03) 0.94 (0.03) |
| $R^2$ | 0.19 (0.14) / 0.12 (0.06) | 0.17 (0.15) / 0.03 (0.18) | 0.19 (0.14) -0.46 (0.46) | 0.18 (0.15) / -1.41 (0.48) |

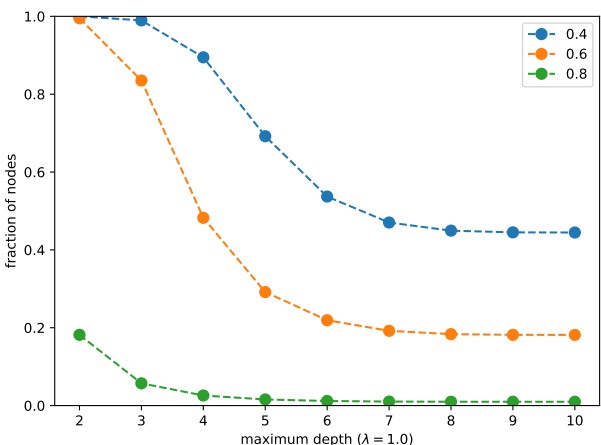

Figure 1: Efficacy of tree pruning using synthetic data.

## 5 CONCLUSION

This paper introduces Huberized-SHIM, an extension of the Sparse High-Order Interaction Model (SHIM) that enhances robustness against outliers while maintaining interpretability in high-stakes applications. The proposed homotopy-based regularization path algorithm and tree-pruning criterion efficiently manage computational complexity, making SHIM scalable for real-world datasets. Additionally, the incorporation of conformal prediction provides statistical coverage guarantees, reinforcing model reliability. Our experiments demonstrate that Huberized-SHIM surpasses standard SHIM in robustness and predictive accuracy, offering a powerful tool for transparent, data-driven decision-making.

## Acknowledgements

D.D. is supported by JSPS KAKENHI 23K16942. K.T. is supported by JST MIRAI, JST ERATO JPMJER1903 and JST CREST JPMJCR21O2.

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

# Statistically Robust Sparse High-order Interaction Model
## (Supplementary Material)

**Diptesh Das**[*1]          **Ichiro Takeuchi**[2, 3]          **Koji Tsuda**[†1, 3, 4]

[1]Department of Computational Biology and Medical Sciences, The University of Tokyo, Japan.
[2]Department of Mechanical Systems Engineering, Nagoya University, Japan.
[3]RIKEN Center for Advanced Intelligence Project, Japan.
[4]Center for Basic Research on Materials, National Institute for Materials Science, Japan.

## A  EXPERIMENTAL SET UP

We considered the following settings in all experiments. We set the $\ell_1$ regularization hyperparameter $\lambda = 1.0$, $\ell_2$ regularization hyperparameter $\alpha = 0.001$, and huber hyperparameter $\delta = 1.0$. We have chosen a random sample of size $n = 300$ which is first (randomly) split into a training set and test test using standard scikit-learn' train_test_split method considering a split-ratio of $= 0.25$. The training set is further split into a proper training set and a calibration using the same train_test_split method considering a split-ratio of $= 0.5$. All experiments are repeated for 10 independent runs and the mean and standard deviations of all 10 independent runs have been reported. We set fixed values of hyper parameters in all our experiments just for the demonstration purpose. However, we recommend tuning the hyper parameters for each data separately using cross validation methods for best model performance. To simulate the effect of outliers we used the following strategy.

```
outlier_indices = np.random.choice(range(
    y_train.shape[0]), n_out, replace=True)
y_train[outlier_indices] += 2*(y_train.max
    () - y_train.min(),)
```

where n_out represents the number of outliers.

**Synthetic data.**  We randomly generated i.i.d. samples $(Z_i, y_i) \in \{0,1\}^m \times \mathbb{R}$, where $i \in [n]$, ensuring that, on average, $100m(1 - \zeta)\%$ of the features in $Z_i \in \mathbb{R}^m$ take a value of 1. The parameter $\zeta \in [0, 1]$ controls the sparsity of the design matrix, while the regularization parameter $\lambda$ governs the sparsity of the model coefficients. The effectiveness of the tree pruning condition relies on the sparsity of the design matrix, leveraging the tree's anti-monotonicity property. Since high-dimensional real-world data tend to be sparse, the choice of $\zeta$ in our experiments serves purely a demonstration purpose. The response

variable $y_i \in \mathbb{R}$ is sampled from the normal distribution $\mathcal{N}(\mu(Z_i), \sigma^2)$. For demonstration, we adopt a true model incorporating up to fourth-order interactions, defined as: $\mu(Z_i) = -z_{i2} + z_{i3} + 20z_{i5} - 7z_{i2}z_{i3}z_{i4} - 20z_{i1}z_{i2}z_{i3}z_{i4}$, where $\sigma = 1$. We set $\zeta = 0.2, m = 10$ for the statistical results in Table 1. To demonstrate the efficacy of tree pruning we used the same true model and $m = 10$, but varied the sparsity level $\zeta \in \{0.4, 0.6, 0.8\}$. This model is used solely for illustrative purposes, and the proposed method is applicable to any chosen model.

**Real data.**  For the real data we considered Entacmaea quadricolor fluorescent protein $eqFP611$, two variant of which namely one bright deep-red ($mKate2, \lambda_{ex} = 590nm, \lambda_{em} = 635nm$) and one bright blue ($mTagBFP2, \lambda_{ex} = 405nm, \lambda_{em} = 460nm$) are separated by thirteen mutations [Poelwijk et al., 2019]. Form biological perspective it is important to identify the crucial mutations and their pattern of epistasis (high-order interactions among mutations) that relate to the phenotypes (e.g., brightness). We also evaluated our approach using ProPublica's COMPAS recidivism dataset, which includes seven categorical and integer-valued features along with continuous recidivism scores. An equivalent set of 14 binary features and continuous response was obtained from the CORELS GitHub repository [Angelino et al., 2017]. Model interpretabilty is crucial for the analysis of such high-stake decision making problems where an algorithm derived predictions are associated with the life of a human being or critical biological analysis. We provide additional results here in addition to the results provided in the main paper. Figure 2 shows benchmark results comparing different regression methods using synthetic SHIM data. For this benchmark results we used the default parameter settings of Scikit-learn implementation of the standard methods. Figure 3 shows the computation time of running Huberized-SHIM with or without the pruning condition using the synthetic SHIM data at different data sparsity levels. Figure 4 and Figure 5 show the efficacy of the proposed tree pruning condition when applied on real-world

---
[*]Corresponding author: diptesh.das@edu.k.u-tokyo.ac.jp
[†]Corresponding author: tsuda@k.u-tokyo.ac.jp

*Accepted for the 8th Workshop on Tractable Probabilistic Modeling at UAI (TPM 2025).*

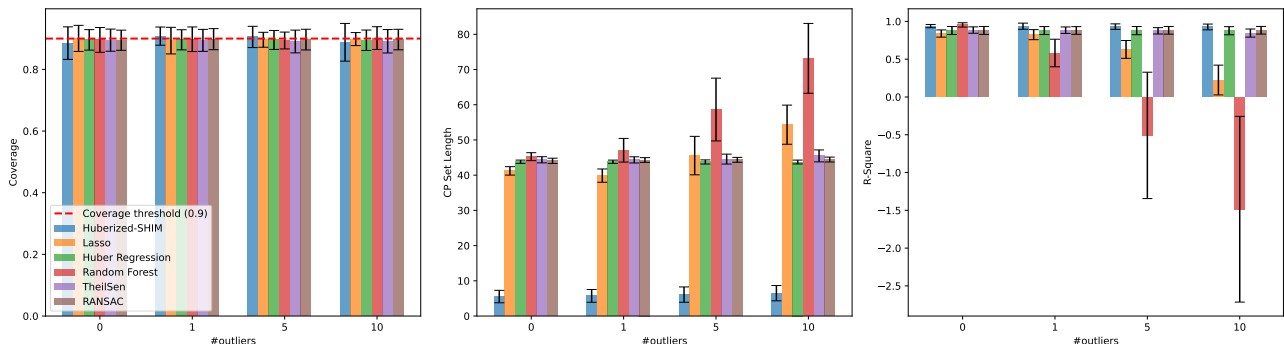

Figure 2: Benchmark results comparing different regression methods using synthetic SHIM data.

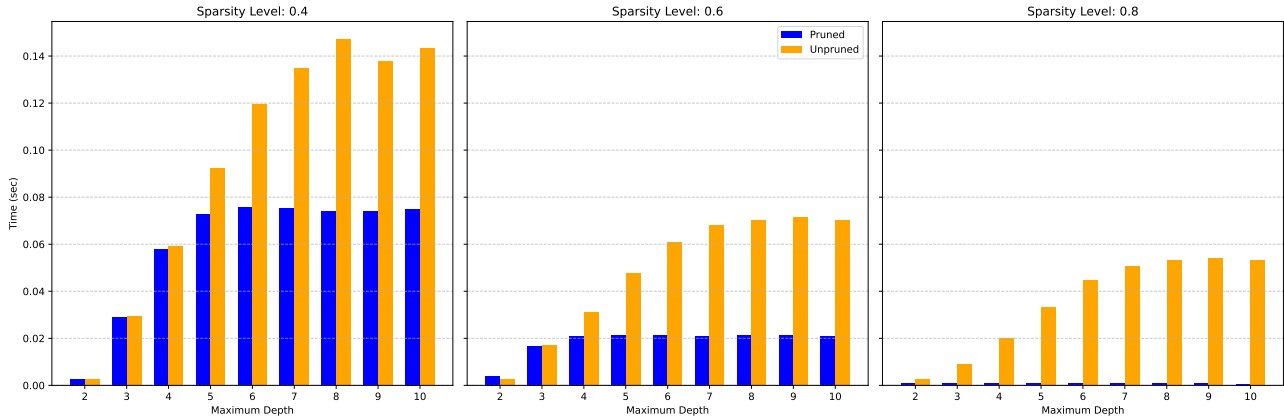

Figure 3: Efficacy of tree pruning using synthetic SHIM data. Computation time (in sec) with and without pruning using three different sparsity levels $\zeta \in \{0.4, 0.6, 0.8\}$. We ran all the experiments on an Apple MacBook Air with an Apple M2 chip (8-cores) and 8 GB memory.

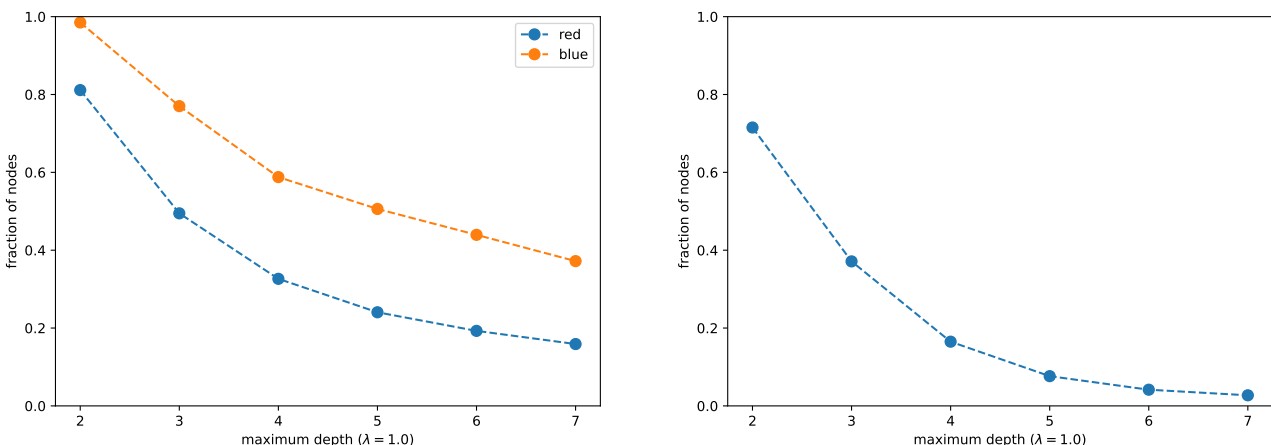

Figure 4: Efficacy of tree pruning using fluorescence data.

Figure 5: Efficacy of tree pruning using compas data.

fluorescence [Poelwijk et al., 2019] and compas data respectively [Angelino et al., 2017]. Table 4 shows the results using blue brightness as the fitness score (response variable).

Table 4: Results using Fluorescence data (fitness='blue').

| | Clean | One outlier | Five outliers | Ten outliers |
|---|---|---|---|---|
| length | 1.17 (0.20) / 1.15 (0.19) | 1.14 (0.18) / 1.24 (0.27) | 1.68 (0.45) / 1.99 (0.33) | 2.26 (0.51) / 2.61 (0.45) |
| cov | 0.90 (0.05) / 0.90 (0.05) | 0.90 (0.06) / 0.88 (0.08) | 0.91 (0.05) / 0.91 (0.03) | 0.91 (0.03) / 0.90 (0.05) |
| $R^2$ | 0.18 (0.13) / 0.18 (0.13) | 0.15 (0.14) / 0.00 (0.17) | -0.53 (0.80) / -1.19 (0.70) | -2.02 (1.43) / -3.21 (1.78) |

# B    TECHNICAL APPENDIX

## B.1    PROOF OF PROPOSITION 1

Before proving Proposition 1, we state the following proposition.

**Proposition 2.** *For any $\lambda \in [\lambda_1, \lambda_2)$, if there is no "knot-crossing" from either side (left or right) for any instance $i \in [n]$, then the values of $a(\lambda)$ and $(1 - a(\lambda))s(r(\lambda))$ remain the same for all $\lambda \in [\lambda_1, \lambda_2)$.*

*Proof of Proposition 2.* If for any $\lambda \in [\lambda_1, \lambda_2)$ there is no "knot-crossing" from either side (left or right), then, $\forall i' \in [n] : |r_{i'}(\lambda_1)| > \delta \implies a_{i'}(\lambda_1) = a_{i'}(\lambda) = 0$ and $(1 - a_{i'}(\lambda_1))s(r_{i'}(\lambda_1)) = s(r_{i'}(\lambda_1)) = s(r_{i'}(\lambda)) \in \{-1, +1\}$, and $\forall i \in [n] : |r_i(\lambda_1)| \leq \delta \implies a_i(\lambda_1) = a_i(\lambda) = 1$, and $(1 - a_i(\lambda))s(r_i(\lambda)) = 0$. Therefore, if there is no "knot-crossing" from either side (left or right) for any instance $i \in [n]$, then the values of $a(\lambda)$ and $(1 - a(\lambda))s(r(\lambda))$ remain the same for all $\lambda \in [\lambda_1, \lambda_2)$.    □

*Proof of Proposition 1.* For any $\lambda \in [\lambda_1, \lambda_2)$, if the $sign(\beta(\lambda))$ remains the same, then the active set also remains the same, that is $\mathcal{A}_\lambda = \mathcal{A}_{\lambda_1}$. Furthermore, we assume that there is no "knot-crossing" for any $\lambda \in [\lambda_1, \lambda_2)$. Then, using Proposition 2 the values of $a(\lambda)$ and $(1 - a(\lambda))s(r(\lambda))$ remain the same for all $\lambda \in [\lambda_1, \lambda_2)$. Now, using optimality condition (4) at $\lambda_1$ and $\lambda_2$, we can write the following:

$$X^\top\Big(a(\lambda_1) \odot r(\lambda_1) + \delta(1 - a(\lambda_1)) \odot s(r(\lambda_1))\Big)$$
$$= \lambda_1 s(\beta(\lambda_1)) + \alpha\beta(\lambda_1), \quad (10)$$
$$X^\top\Big(a(\lambda_2) \odot r(\lambda_2) + \delta(1 - a(\lambda_2)) \odot s(r(\lambda_2))\Big)$$
$$= \lambda_2 s(\beta(\lambda_2)) + \alpha\beta(\lambda_2). \quad (11)$$

Therefore, subtracting (10) from (11), expanding $r(\lambda) = y - X_{\mathcal{A}_\lambda}\beta_{\mathcal{A}_\lambda}(\lambda)$ and using Proposition 2 we can write for the active components of the optimum solutions $(\beta_{\mathcal{A}_\lambda}(\lambda))$:

$$-X_{\mathcal{A}_\lambda}^\top(a(\lambda) \odot X_{\mathcal{A}_\lambda})\big(\beta_{\mathcal{A}_\lambda}(\lambda_2) - \beta_{\mathcal{A}_\lambda}(\lambda_1)\big)$$
$$= \alpha\big(\beta_{\mathcal{A}_\lambda}(\lambda_2) - \beta_{\mathcal{A}_\lambda}(\lambda_1)\big) + (\lambda_2 - \lambda_1)s\big(\beta_{\mathcal{A}_\lambda}(\lambda)\big),$$
$$(12)$$

$$\Big(\alpha\mathbb{I}_{|\mathcal{A}_\lambda|} + X_{\mathcal{A}_\lambda}^\top(a(\lambda) \odot X_{\mathcal{A}_\lambda})\Big)\big(\beta_{\mathcal{A}_\lambda}(\lambda_2) - \beta_{\mathcal{A}_\lambda}(\lambda_1)\big)$$
$$= (\lambda_1 - \lambda_2)s\big(\beta_{\mathcal{A}_\lambda}(\lambda)\big),$$

$$\implies \frac{\big(\beta_{\mathcal{A}_\lambda}(\lambda_2) - \beta_{\mathcal{A}_\lambda}(\lambda_1)\big)}{\lambda_1 - \lambda_2}$$
$$= \Big(\alpha\mathbb{I}_{|\mathcal{A}_\lambda|} + X_{\mathcal{A}_\lambda}^\top(a(\lambda) \odot X_{\mathcal{A}_\lambda})\Big)^{-1} s\big(\beta_{\mathcal{A}_\lambda}(\lambda)\big),$$
$$= \psi_{\mathcal{A}_\lambda}(\lambda).$$

Similarly, for the non-active components of the optimum solutions $(\beta_{\mathcal{A}_\lambda^c}(\lambda))$, we can write the following:

$$-X_{\mathcal{A}_\lambda^c}^\top(a(\lambda) \odot X_{\mathcal{A}_\lambda})\big(\beta_{\mathcal{A}_\lambda}(\lambda_2) - \beta_{\mathcal{A}_\lambda}(\lambda_1)\big) \quad (13)$$
$$= \lambda_2 s\big(\beta_{\mathcal{A}_\lambda^c}(\lambda_2)\big) - \lambda_1 s\big(\beta_{\mathcal{A}_\lambda^c}(\lambda_1)\big),$$

$$-X_{\mathcal{A}_\lambda^c}^\top(a(\lambda) \odot X_{\mathcal{A}_\lambda})\psi_{\mathcal{A}_\lambda}(\lambda)(\lambda_1 - \lambda_2)$$
$$= \lambda_2 s\big(\beta_{\mathcal{A}_\lambda^c}(\lambda_2)\big) - \lambda_1 s\big(\beta_{\mathcal{A}_\lambda^c}(\lambda_1)\big),$$

$$\implies \frac{\lambda_2 s\big(\beta_{\mathcal{A}_\lambda^c}(\lambda_2)\big) - \lambda_1 s\big(\beta_{\mathcal{A}_\lambda^c}(\lambda_1)\big)}{\lambda_1 - \lambda_2}$$
$$= -X_{\mathcal{A}_\lambda^c}^\top(a(\lambda) \odot X_{\mathcal{A}_\lambda})\psi_{\mathcal{A}_\lambda}(\lambda),$$
$$= \gamma_{\mathcal{A}_\lambda^c}(\lambda).$$
$$\square$$

## B.2    DERIVATION OF STEP-SIZE OF INCLUSION.

At any $\lambda_{t+1} : \lambda_t > \lambda_{t+1} > 0$, any $j \in \mathcal{A}_{\lambda_t}^c$ becomes active if for any $k \in \mathcal{A}_{\lambda_t}$ the following condition is satisfied:

$$|x_j^\top h(\lambda_{t+1}) - \alpha\beta_j(\lambda_{t+1})| = |x_k^\top h(\lambda_{t+1}) - \alpha\beta_k(\lambda_{t+1})|. \quad (14)$$

Now, considering $\beta_k(\lambda_{t+1}) = \beta_k(\lambda_t) + \Delta_1(j) \cdot \psi_k(\lambda_t)$, $v(\lambda_t) = X_{\mathcal{A}_{\lambda_t}}\psi_{\mathcal{A}_{\lambda_t}}$ and expanding $h(\lambda_{t+1})$, where $r(\lambda_{t+1}) = r(\lambda_t) - \Delta_1(j) \cdot v(\lambda_t)$ and and assuming that there is no "knot-crossing" (Proposition 2) for any $\lambda \in [\lambda_t, \lambda_{t+1}]$ such that $a(\lambda_{t+1}) = a(\lambda_t)$ and $(1 - a(\lambda_{t+1}))s(r(\lambda_{t+1})) = (1 - a(\lambda_t))s(r(\lambda_t))$, we can write for the positive terms of equicorrelation condition (14):

$$x_j^\top\big(a(\lambda_t) \odot \big(r(\lambda_t) - \Delta_1^+(j) \cdot v(\lambda_t)\big)$$
$$+ \delta(1 - a(\lambda_t)) \odot s(r(\lambda_t))\big)$$
$$= x_k^\top\big(a(\lambda_t) \odot \big(r(\lambda_t) - \Delta_1^+(j) \cdot v(\lambda_t)\big)$$
$$+ \delta(1 - a(\lambda_t)) \odot s(r(\lambda_t))\big)$$
$$- \alpha\big(\beta_k(\lambda_t) + \Delta_1^+(j) \cdot \psi_k(\lambda_t)\big),$$

$$\implies (x_j - x_k)^\top h(\lambda_t) + \alpha\beta_k(\lambda_t)$$
$$= \Delta_1^+(j)((x_j - x_k)^\top(a(\lambda_t) \odot v(\lambda_t)) - \alpha\psi_k(\lambda_t)),$$

$$\implies \Delta_1^+(j) = \frac{(x_j - x_k)^\top h(\lambda_t) + \alpha\beta_k(\lambda_t)}{(x_j - x_k)^\top(a(\lambda_t) \odot v(\lambda_t)) - \alpha\psi_k(\lambda_t)}. \quad (15)$$

Similarly for the negative terms of equicorrelation (19), we can write

$$\Delta_1^-(j) = \frac{(x_j + x_k)^\top h(\lambda_t) - \alpha\beta_k(\lambda_t)}{(x_j + x_k)^\top (a(\lambda_t) \odot v(\lambda_t)) + \alpha\psi_k(\lambda_t)}. \quad (16)$$

Now, combining (15) and (16), we can write the step-size of inclusion

$$\Delta_1(j) = \frac{(x_j \mp x_k)^\top h(\lambda_t) \pm \alpha\beta_k(\lambda_t)}{(x_j \mp x_k)^\top (a(\lambda_t) \odot v(\lambda_t)) \mp \alpha\psi_k(\lambda_t)}. \quad (17)$$

### B.3 PROOF OF TREE PRUNING CONDITION

Before proving Lemma 1, we define few definitions and two propositions as stated below.

**Definition 1.** *A tree is constructed in such a way that for any pair of nodes $(\ell, \ell')$, where $\ell$ is the ancestor of $\ell'$, that is $\ell \subset \ell'$, the following conditions are satisfied $\forall i \in [n]$:*

$$x_{i\ell'} = 1 \implies x_{i\ell} = 1 \quad and \quad x_{i\ell} = 0 \implies x_{i\ell'} = 0.$$

Let's define $\forall \ell \in [p]$,

$$\begin{aligned}
\rho_\ell(\lambda) &= x_\ell^\top w(\lambda), \\
\eta_\ell(\lambda) &= x_\ell^\top u(\lambda), \\
\theta_\ell(\lambda) &= x_\ell^\top \kappa(\lambda),
\end{aligned} \quad (18)$$

where $w(\lambda) = a(\lambda) \odot r(\lambda)$, $u(\lambda) = a(\lambda) \odot v(\lambda)$, $v(\lambda) = X\psi(\lambda)$, and $\kappa(\lambda) = \delta(1 - a(\lambda)) \odot s(r(\lambda))$.

**Proposition 3.** *Let's define for a vector $g \in \mathbb{R}^n$*

$$b_\ell(g) = \max\left\{\sum_{g_i<0}|g_i|x_{i\ell}, \sum_{g_i>0}|g_i|x_{i\ell}\right\},$$

*then if we expand $\rho_\ell(\lambda), \eta_\ell(\lambda)$ and $\theta_\ell(\lambda)$ separately for positive and negative values of $w_i(\lambda), u_i(\lambda)$ and $\kappa_i(\lambda), \forall i \in [n]$ respectively, we can write*

$$\begin{aligned}
|\rho_\ell(\lambda)| &\leq b_\ell(w(\lambda)), \\
|\eta_\ell(\lambda)| &\leq b_\ell(u(\lambda)), \\
|\theta_\ell(\lambda)| &\leq b_\ell(\kappa(\lambda))
\end{aligned}$$

*Proof of Proposition 3.* We have

$$\begin{aligned}
|x_\ell^\top g| &= \left|\sum_{i=1}^n g_i x_{i\ell}\right| \\
&= \left|\sum_{g_i>0}|g_i|x_{i\ell} - \sum_{g_i<0}|g_i|x_{i\ell}\right| \\
&\leq \max\left\{\sum_{g_i>0}|g_i|x_{i\ell}, \sum_{g_i<0}|g_i|x_{i\ell}\right\} \\
&=: b_\ell(g).
\end{aligned}$$

$\square$

Here, we used a generic vector $g$ in place of $w(\lambda), u(\lambda)$ and $\kappa(\lambda)$ to keep the proof simple.

**Proposition 4.** *By using the tree anti-monotonicity property i.e., $x_{i\ell} \geq x_{i\ell'}, \forall \ell' \supset \ell, \forall i \in [n]$ as defined in the definition of tree (Definition 1), we have*

$$b_\ell(g) \geq b_{\ell'}(g).$$

*Proof of Proposition 4.* From the definition of tree we have $x_{i\ell} \geq x_{i\ell'}, \forall \ell' \supset \ell, \forall i \in [n]$. Hence, we can write

$$\begin{aligned}
b_\ell(g) &= \max\left\{\sum_{g_i<0}|g_i|x_{i\ell}, \sum_{g_i>0}|g_i|x_{i\ell}\right\} \\
&\geq \max\left\{\sum_{g_i<0}|g_i|x_{i\ell'}, \sum_{g_i>0}|g_i|x_{i\ell'}\right\} \\
&=: b_{\ell'}(g).
\end{aligned}$$

$\square$

**Feature inclusion condition.** At any $\lambda_2 : \lambda_1 > \lambda_2 > 0$, any $j \in \mathcal{A}_{\lambda_1}^c$ becomes active if for any $k \in \mathcal{A}_{\lambda_1}$ the following equicorrelation condition is satisfied:

$$\begin{aligned}
&\left|x_j^\top\left(a(\lambda_2) \odot r(\lambda_2) + \delta(1 - a(\lambda_2)) \odot s(r(\lambda_2))\right)\right. \\
&\qquad\qquad \left. - \alpha\beta_j(\lambda_2)\right| \\
&= \left|x_k^\top\left(a(\lambda_2) \odot r(\lambda_2) + \delta(1 - a(\lambda_2)) \odot s(r(\lambda_2))\right)\right. \\
&\qquad\qquad \left. - \alpha\beta_k(\lambda_2)\right|. \quad (19)
\end{aligned}$$

The above equicorrelation condition (19) can be used to derive the tree-pruning condition. If the active set $\mathcal{A}_\lambda$ and the sign of the model coefficients $s(\beta_j(\lambda)), \forall j \in [p]$ remain the same and also there exists no "knot-crossing" (Proposition 2) such that $a_i(\lambda), (1-a_i(\lambda))s(r_i(\lambda)), \forall i \in [n]$ remain constant for $\forall \lambda \in [\lambda_1, \lambda_2]$, then (19) can be rewritten as follows:

$$\begin{aligned}
&\left|x_j^\top\left(a(\lambda_1) \odot r(\lambda_2) + \delta(1 - a(\lambda_1)) \odot s(r(\lambda_1))\right)\right. \\
&\qquad\qquad \left. - \alpha\beta_j(\lambda_2)\right| \\
&= \left|x_k^\top\left(a(\lambda_1) \odot r(\lambda_2) + \delta(1 - a(\lambda_1)) \odot s(r(\lambda_1))\right)\right. \\
&\qquad\qquad \left. - \alpha\beta_k(\lambda_2)\right|, \quad (20)
\end{aligned}$$

Then, expanding $r(\lambda_2) = r(\lambda_1) - \Delta_1(j)v(\lambda_1)$ and using the definitions in (18), we can rewrite (20):

$$\begin{aligned}
&\left|\rho_j(\lambda_1) - \Delta_1(j)\eta_j(\lambda_1) + \theta_j(\lambda_1) - \alpha\big(\beta_j(\lambda_1)\right. \\
&\qquad\qquad \left. + \Delta_1(j) \cdot \psi_j(\lambda_1)\big)\right| \\
&= \left|\rho_k(\lambda_1) - \Delta_1(j)\eta_k(\lambda_1) + \theta_k(\lambda_1) - \alpha\big(\beta_k(\lambda_1)\right. \\
&\qquad\qquad \left. + \Delta_1(j) \cdot \psi_k(\lambda_1)\big)\right|, \quad (21)
\end{aligned}$$

$$\implies |\rho_j(\lambda_1) - \Delta_1(j)\eta_j(\lambda_1) + \theta_j(\lambda_1)|$$
$$= |\rho_k(\lambda_1) - \alpha\beta_k(\lambda_1) + \theta_k(\lambda_1) - \Delta_1(j)(\eta_k(\lambda_1)$$
$$+ \alpha\psi_k(\lambda_1))|. \quad (22)$$

We derived (22) from (21), by considering the fact that $\beta_j(\lambda) = 0$ and $\psi_j(\lambda) = 0, \forall j \in \mathcal{A}_\lambda^c$.

**Tree pruning (Branch & Bound).** To have a solution of (22), the following conditions must be satisfied:

$$|\rho_j(\lambda_1) - \Delta_1(j)\eta_j(\lambda_1) + \theta_j(\lambda_1)|$$
$$\leq |\rho_j(\lambda_1)| + \Delta_1(j)|\eta_j(\lambda_1)| + |\theta_j(\lambda_1)|, \quad (23)$$

and

$$|\rho_k(\lambda_1) - \alpha\beta_k(\lambda_1) + \theta_k(\lambda_1) - \Delta_1(j)(\eta_k(\lambda_1) + \alpha\psi_k(\lambda_1))|$$
$$\geq |\rho_k(\lambda_1) - \alpha\beta_k(\lambda_1)| - \Delta_1(j)|\eta_k(\lambda_1) \quad (24)$$
$$+ \alpha\psi_k(\lambda_1)| - |\theta_k(\lambda_1)|.$$

Therefore, (23) and (24) implies

$$|\rho_j(\lambda_1)| + \Delta_1(j)|\eta_j(\lambda_1)| + |\theta_j(\lambda_1)|$$
$$\geq |\rho_k(\lambda_1) - \alpha\beta_k(\lambda_1)| - \Delta_1(j)|\eta_k(\lambda_1)$$
$$+ \alpha\psi_k(\lambda_1)| - |\theta_k(\lambda_1)|. \quad (25)$$

Therefore, if the condition stated in (26) is satisfied, then there will be no solution of (22), and hence, (26) can be used to derive a tree pruning condition.

$$|\rho_j(\lambda_1)| + \Delta_1(j)|\eta_j(\lambda_1)| + |\theta_j(\lambda_1)|$$
$$< |\rho_k(\lambda_1) - \alpha\beta_k(\lambda_1)| - \Delta_1(j)|\eta_k(\lambda_1)$$
$$+ \alpha\psi_k(\lambda_1)| - |\theta_k(\lambda_1)|. \quad (26)$$

By using Proposition 3, we can write (27) which implies (26).

$$b_j(w(\lambda_1)) + \Delta_1(j)b_j(u(\lambda_1)) + b_j(\kappa(\lambda_1))$$
$$< |\bar{\rho}_k(\lambda_1)| - \Delta_1(j)|\bar{\eta}_k(\lambda_1)| - |\theta_k(\lambda_1)|. \quad (27)$$

where $\bar{\rho}_k(\lambda) = \rho_k(\lambda) - \alpha\beta_k(\lambda), \bar{\eta}_k(\lambda) = \eta_k(\lambda) + \alpha\psi_k(\lambda)$.

*Proof of Lemma 1.* We now prove Lemma 1 by contradiction, that is we assume that at any node $\ell$, the condition (9) stated in Lemma 1 holds, and there exists one $\ell' \supset \ell$ : $\Delta_1(\ell') < \Delta_1(\ell_1^\dagger)$; then show that this is a contradiction.

Therefore $\quad |\bar{\rho}_k(\lambda)| - \Delta_1(\ell')|\bar{\eta}_k(\lambda)| - |\theta_k(\lambda)|$
$$> |\bar{\rho}_k(\lambda)| - \Delta_1(\ell_1^\dagger)|\bar{\eta}_k(\lambda)| - |\theta_k(\lambda)|,$$
$$\text{because } \Delta_1(\ell') < \Delta_1(\ell_1^\dagger)$$
$$> b_\ell(w(\lambda)) + \Delta_1(\ell_1^\dagger)b_\ell(u(\lambda)) + b_\ell(\kappa(\lambda)),$$
$$\text{using (9)}$$
$$> b_{\ell'}(w(\lambda)) + \Delta_1(\ell_1^\dagger)b_{\ell'}(u(\lambda)) + b_{\ell'}(\kappa(\lambda)),$$
$$\text{(Proposition 4),}$$
$$> b_{\ell'}(w(\lambda)) + \Delta_1(\ell')b_{\ell'}(u(\lambda)) + b_{\ell'}(\kappa(\lambda)),$$
$$\text{because } \Delta_1(\ell') < \Delta_1(\ell_2^\dagger).$$

Therefore, we got

$$|\bar{\rho}_k(\lambda)| - \Delta_1(\ell')|\bar{\eta}_k(\lambda)| - |\theta_k(\lambda)|$$
$$> b_{\ell'}(w(\lambda)) + \Delta_1(\ell')b_{\ell'}(u(\lambda)) + b_{\ell'}(\kappa(\lambda))$$
$$\implies \ell' \text{ is infeasible,}$$
$$\text{(using (27))}$$
$$\implies \Delta_1(\ell') \geq \Delta_1(\ell_1^\dagger).$$

$\square$

This completes the proof of Lemma 1. Hence, if the pruning condition in Lemma 1 holds, then we do not need to search the sub-tree with $\ell$ as the root node, and hence increasing the efficiency of the search procedure.

### B.4 ALGORITHM OF HUBERIZED-SHIM.

**Derivation of first $\ell^*$ and $\lambda_{max}$ :** Let's define

$$G(\ell) = |X_\ell^\top h(\lambda)|, \quad \text{then}$$

$$\ell^* = \arg\max_{\ell \in [p]} G(\ell),$$

$$\lambda_{max} = G(\ell^*), \quad (28)$$

---

**Algorithm 1** Exact $\lambda$-path of Huberized SHIM
1: **Input:** $\mathcal{D}_n = \{(X_i, y_i)\}_{i=1}^n$
2: Initialize $t = 0$, $\lambda_0 = \lambda_{max}$ using (28), $\mathcal{A}_{\lambda_0} = \{\ell^*\}$, $a_i(\lambda_0)$ and $s(r_i(\lambda_0))$ using (3)
3: **while** $(\lambda > 0)$ **do**
4:     Compute $\Delta$ using (8)
5:     Update: $\lambda_{t+1} \leftarrow \lambda_t + \Delta$, $\beta_{\mathcal{A}_{\lambda_t}}(\lambda_{t+1}) \leftarrow \beta_{\mathcal{A}_{\lambda_t}}(\lambda_t) + \Delta \cdot \psi_{\mathcal{A}_{\lambda_t}}(\lambda_t)$, $\beta_{t+1} \leftarrow [\beta_{\mathcal{A}_{\lambda_t}}(\lambda_{t+1}), \mathbf{0}]$
6:     **if** $\Delta = \Delta_{\lambda_1}$ **then**
7:         add $\ell$ into $\mathcal{A}_{\lambda_t}$
8:     **else if** $\Delta = \Delta_{\lambda_2}$ **then**
9:         remove $\ell$ from $\mathcal{A}_{\lambda_t}$
10:    **else if** $\Delta = \Delta_{\lambda_3}$ **then**
11:        update $a_i(\lambda), \forall i \in [n]$ using (3)
12:    **end if**
13:    $\mathbb{A} = \mathbb{A} \cup \mathcal{A}(\lambda_{t+1}), \mathbb{B} = \mathbb{B} \cup \{\beta_{t+1}\}$
14:    Update $\psi_{\mathcal{A}_{\lambda_t}}(\lambda_t)$ using (7)
15:    t = t +1
16: **end while**
17: **Output:** $\mathbb{A}, \mathbb{B}$

---

## C CONFORMAL PREDICTION

A single point estimate is inadequate for automated decision-making in high-risk domains like medical diagnosis and criminal justice [Angelino et al., 2018, Rudin, 2019, Das

et al., 2019]. In such critical scenarios, equipping estimators with coverage information enhances decision-makers' confidence, enabling more informed and reliable choices when stakes are high. Given a labelled dataset $\mathcal{D}_n = \{(x_i, y_i)\}_{i=1}^n$ and a new observation $x_{n+1}$, the objective of the conformal prediction (CP) framework is to generate a statistically valid prediction set $\mathcal{C}(x_{n+1})$ for the unknown response $y_{n+1}$, ensuring coverage guarantees[Vovk et al., 2005, Shafer and Vovk, 2008], i.e.,

$$\mathbb{P}(y_{n+1} \in \mathcal{C}(x_{n+1})) \geq 1 - \alpha, \qquad (29)$$

where $\alpha \in [0, 1]$ determines the level of coverage. In inductive conformal prediction [Papadopoulos et al., 2002, Angelopoulos and Bates, 2021], also known as split-CP, the dataset $\mathcal{D}_n$ is divided into two distinct subsets: the training set $\mathcal{D}_{\text{tr}} = \{(x_1, y_1), \ldots, (x_{n'}, y_{n'})\}$ and the calibration set $\mathcal{D}_{\text{cal}} = \{(x_{n'+1}, y_{n'+1}), \ldots, (x_n, y_n)\}$, where $n' < n$. A regression model $\mu^{\text{tr}}(\cdot)$ is trained using the training set $\mathcal{D}_{\text{tr}}$ only once, and the $p$-values, represented as $\pi_{\text{split}}(\cdot)$, are subsequently computed based on the calibration set $\mathcal{D}_{\text{cal}}$ :

$$\pi_{\text{split}}(\tau) = 1 - \frac{1}{n - n'} \sum_{i=n'+1}^{n} \mathbb{1}_{S_i^{\text{cal}}(\tau) \leq S_{n+1}(\tau)},$$

where $S_i^{\text{cal}}(\tau) = |y_i - \mu^{\text{tr}}(x_i)|, \forall i \in [n' + 1, n]$. Therefore, the split-CP set can be defined as follows:

$$\mathcal{C}_{\text{split}}(x_{n+1}) = \{\tau : \pi_{\text{split}}(\tau) \geq \alpha\}.$$

When the conformity score is defined as the absolute residual, then the split-CP set can be conveniently written as $\mathcal{C}_{\text{split}}(x_{n+1}) = [\mu^{\text{tr}}(x_{n+1}) \pm Q_{1-\alpha}^{\text{cal}}]$, where $Q_{1-\alpha}^{\text{cal}}$ is the $(1 - \alpha)$ quantile of the calibration scores $S_i^{\text{cal}}, \forall i \in [n' + 1, n]$. For more details about conformal prediction please see [Shafer and Vovk, 2008, Angelopoulos and Bates, 2021].