# OpenReview forum: "Statistically Robust Sparse High-order Interaction Model"
_auai.org/UAI/2025/Workshop/TPM — TPM 2025_

### Official Review · Reviewer_vTck · 2025-06-14
**Efficient SHIMs with Huber Loss, Regularisation Paths, and Tree Pruning**

**Rating:** 2

**Review:**

This paper introduces Huberised-SHIM, a robust extension of Sparse High-Order Interaction Models (SHIMs) that replaces the squared loss with Huber loss to reduce sensitivity to outliers. To manage the combinatorial complexity of high-order interactions, the authors propose a tree-pruning strategy integrated into a homotopy method for computing the piecewise-linear regularisation path for the $\ell_1$ penalty parameter $\lambda$. The method is evaluated on synthetic and real-world datasets, with robustness assessed via inductive conformal prediction.

Due to limited familiarity with homotopy methods and the SHIM framework, I could not verify the theoretical correctness of the derivations.

**Key Contributions**

- Incorporation of Huber loss into SHIM for improved robustness to outliers.
- A homotopy algorithm for efficiently computing the regularisation path for $\lambda$.
- A new pruning criterion to efficiently construct interaction trees, especially effective under high sparsity.
- Empirical evaluation using inductive conformal prediction, demonstrating improved robustness on synthetic and real-world datasets.

**Weaknesses**

- The pruning and homotopy methods are technically detailed and would benefit from a higher-level, perhaps graphical, intuition. Space could be made by shortening the extensive discussion of application areas in the introduction. The mathematical notation is often dense and occasionally under-defined, making the paper harder to follow for non-experts. For example:
  - Below Eq. (1), $E(e|X) = 0$ is used, but $X$ is not yet defined.
  - Below Eq. (29), $S_{n+1}(\tau)$ is used without explanation.
- The empirical evaluation is limited in scale (up to 14, mostly binary, input features; 300 samples; 3rd-order interactions), and the main text omits runtime results (briefly reported as max 0.08s in the appendix). There is no discussion of runtime scalability or comparison to baselines.
- Outlier injection is based on a single setting (adding twice the response range to a subset). A broader range of outlier types and clearer reporting of the outlier-to-sample ratio would strengthen the analysis.
- The regularisation path is not empirically explored; only a fixed $\lambda$ is used. Sensitivity to $\lambda$, the $\ell_2$ regularisation parameter $\alpha$, and the Huber parameter $\delta$ is not discussed.

In conclusion, while the technical contributions are solid, the paper would benefit from a clearer method description and more comprehensive empirical evaluation.

---

### Official Review · Reviewer_ruPY · 2025-06-16
**nice technical contribution**

**Rating:** 3

**Review:**

# Summary
The paper introduces an extension of Sparse High-order Interaction Models (SHIMs) designed to counter the negative effects of outliers on feature revelance attributions.  In a nutshell, outliers can skew the model towards considering as relevant more features/attributes than necessary.  The proposed solution amounts to replacing a (naive) regression loss with a more robust Huber loss. This introduces a scalability challenge, as optimizing the robust loss is non-trivial. The authors solve this by introducing a novel optimization setup based on an analysis of the geometry of the optima of the robust loss for SHIMs, paired with a novel pruning strategy for filtering out poor solutions. The experiments indicate that in a conformal prediction setting, the conformal sets tend to be smaller for robust SHIMs compared to the original model, with no loss in performance for increasing amounts of outliers.
# Strengths

Generally well written and self-contained, with a few typos.
The motivation is quite convincing (and backed up by the empirical results)
The contribution appears to be technically sound (although I did not check the derivations in detail).
The results are promising, in the sense that robust models seem to have an advantage in the tested settings.
The topic is relevant for TPM.
# Weaknesses

The experiments are relatively restricted: robust SHIM is compared against SHIM on three data sets with artificially added outliers, that's it.  For instance, runtime is not discussed, and real-world outliers are not considered.  They are however enough for a workshop contribution.